# Linear Disentangled Representations and Unsupervised Action Estimation

**Matthew Painter, Jonathon Hare, Adam Prügel-Bennett**
Department of Electronics and Computer Science
Southampton University
{mp2u16, jsh2, apb}@ecs.soton.ac.uk

## Abstract

Disentangled representation learning has seen a surge in interest over recent times, generally focusing on new models which optimise one of many disparate disentanglement metrics. Symmetry Based Disentangled Representation learning introduced a robust mathematical framework that defined precisely what is meant by a "linear disentangled representation". This framework determined that such representations would depend on a particular decomposition of the symmetry group acting on the data, showing that actions would manifest through irreducible group representations acting on independent representational subspaces. Caselles-Dupré et al. [2019] subsequently proposed the first model to induce and demonstrate a linear disentangled representation in a VAE model. In this work we empirically show that linear disentangled representations are not generally present in standard VAE models and that they instead require altering the loss landscape to induce them. We proceed to show that such representations are a desirable property with regard to classical disentanglement metrics. Finally we propose a method to induce irreducible representations which forgoes the need for labelled action sequences, as was required by prior work. We explore a number of properties of this method, including the ability to learn from action sequences without knowledge of intermediate states and robustness under visual noise. We also demonstrate that it can successfully learn 4 independent symmetries directly from pixels.

## 1   Introduction

Many learning machines make use of an internal representation [Bengio et al., 2013] to help inform their decisions. It is often desirable for these representations to be interpretable in the sense that we can easily understand how individual parts contribute to solving the task at hand. Interpretable representations have slowly become the major goal in the sub-field of deep learning concerned with Variational Auto-Encoders (VAEs) [Kingma and Welling, 2014], seceding from the usual goal of generative models, accurate/realistic sample generation. In this area, representations generally take the form of a multi-dimensional vector space (latent space), and the particular form of interpretability is known as Disentanglement, where each latent dimension (or group of such) is seen to represent an individual (and independent) generative/explanatory factor of the data. Standard examples of such factors include the x position, the y position, an object's rotation, etc. Effectively separating them into separate subspaces, "disentangling them" is the generally accepted aim of disentanglement research.

Assessing disentangled representations (in the specific case of VAEs) has primarily been based on the application of numerous "disentanglement metrics". These metrics operate on disparate understandings of what constitutes ideal disentanglement, and required large scale work [Locatello et al., 2018] to present correlations between them and solidify the fact that they should be considered jointly and not as individuals. Symmetry based disentangled representation learning (SBDRL) [Higgins et al.,

2018] offers a mathematical framework through which a rigorous definition of a linear disentangled representation can be formed. In essence, if data has a given symmetry structure (expressed through Group Theory), the linear disentangled representations are exactly those which permit irreducible representations of group elements. Caselles-Dupré et al. [2019] propose modelling and inducing such representations through observation of action transitions under the framework of reinforcement learning. Their model successfully induces linear disentangled representations with respect to the chosen symmetry structure, demonstrating that they are achievable through simple methods. Whilst Caselles-Dupré et al. [2019] assume symmetry structure comprising of purely cyclic groups, Quessard et al. [2020] expand this to the more expressive class of SO(3) matrices.

Current work shows that irreducible representations can be induced in latent spaces, but has yet to determine if they can be found without explicitly structuring the loss landscape. Furthermore, they are not related back to previous disentanglement metrics to demonstrate the utility of such structures being present. Finally, by sticking to the reinforcement framework, Caselles-Dupré et al. [2019] and Quessard et al. [2020] allow the model direct knowledge of which actions they are observing. This restricts the applicable domain to data where action transition pairs are explicitly labelled.

In this work, we make the following contributions:

- We confirm empirically that irreducible representations are not generally found in standard VAE models without biasing the loss landscape towards them.
- We determine that inducing such representations in VAE latent spaces garners improved performance on a number of standard disentanglement metrics.
- We introduce a novel disentanglement metric to explicitly measure linear disentangled representations and we modify the mutual information gap metric to be more appropriate for this setting.
- We propose a method to induce irreducible representations without the need for labelled action-transition pairs.
- We demonstrate a number of properties of such a model, show that it can disentangle 4 separate symmetries directly from pixels and show it continues to lead to strong scores on classical disentanglement metrics.

## 2 Symmetry Based Disentangled Representation Learning and Prior Works

This section provides a high level overview of the SBDRL framework without the mathematical grounding in Group and Representation theory on which it is based. The work and appendices of Higgins et al. [2018] offer a concise overview of the necessary definitions and theorems. We encourage the reader to first study their work since they provide intuition and examples which we cannot present here given space constraints.

**SBDRL**   VAE representation learning is concerned with the mapping from an observation space $\mathcal{O} \subset \mathbb{R}^{n_x \times n_y}$ (generally images) to a vector space forming the latent space $\mathcal{Z} \subset \mathbb{R}^l$. SBDRL includes the additional construct of a world space $\mathcal{W} \subset \mathbb{R}^d$ containing the possible states of the world which observations represent. There exists a generative process $b : \mathcal{W} \to \mathcal{O}$ and a inference process $h : \mathcal{O} \to \mathcal{Z}$, the latter being accessible and parametrised by the VAE encoder. SBDRL assumes for convenience that both $h$ and $b$ are injective. All problems in this work have the property $|\mathcal{W}| = |\mathcal{O}| = |\mathcal{Z}| = N$, i.e. there are a finite number of world states and there is no occlusion in observations. We should note however that neither of these are required for our work.

SBDRL proposes to disentangle *symmetries* of the world space, transformations that preserve some (mathematical or physical) feature, often object identity. Specific transformations may be translation of an object or its rotation, both independent of any other motions (note the similarity to generative data factors). Such symmetries are described by symmetry groups and the symmetry structure of the world space is represented by a group with decomposition $G = G_1 \times \cdots \times G_s$ acting on $\mathcal{W}$ through action $\cdot_{\mathcal{W}} : G \times \mathcal{W} \to \mathcal{W}$. The component groups $G_i$ reflect the individual symmetries and the particular decomposition need not be unique.

SBDRL calls $\mathcal{Z}$ disentangled with respect to decomposition $G = G_1 \times \cdots \times G_s$ if:

1. There is a group action $\cdot_{\mathcal{Z}} : G \times \mathcal{Z} \to \mathcal{Z}$

2. The composition $f = h \circ b : \mathcal{W} \to \mathcal{Z}$ is equivariant with respect to the group actions on $\mathcal{W}$ and $\mathcal{Z}$. i.e. $g \cdot_{\mathcal{Z}} f(w) = f(g \cdot_{\mathcal{W}} w) \ \forall w \in \mathcal{W}, g \in G$.

3. There is a decomposition $\mathcal{Z} = \mathcal{Z}_1 \times \cdots \times \mathcal{Z}_s$ such that $\mathcal{Z}_i$ is fixed by the action of all $G_j, j \neq i$ and affected only by $G_i$

Since we assume $\mathcal{Z}$ is a (real) vector space, SBDRL refines this through group representations, constructs which preserve the linear structure. Define a group representation $\rho : G \to GL(V)$ as disentangled with respect to $G = G_1 \times \cdots \times G_s$ if there exists a decomposition $V = V_1 \oplus \cdots \oplus V_s$ and representations $\rho_i : G_i \to GL(V_i)$ such that $\rho = \rho_1 \oplus \cdots \oplus \rho_s$, i.e. $\rho(g_1, \ldots, g_s)(v_1, \ldots, v_s) = (\rho_1(g_1)v(1), \ldots, \rho_s(g_s)v_s)$. A consequence of this is that $\rho$ is disentangled if each factor of $\rho$ is irreducible. Note that we can associate an action with a group representation through $g \cdot w = \rho(g)(w)$.

Given this, a linear disentangled representation is defined in SBDRL to be any $f : \mathcal{W} \to \mathcal{Z}$ that admits a disentangled group representation with respect to the decomposition $G = G_1 \times \cdots \times G_s$. As such we can look for mappings to $\mathcal{Z}$ where actions by $G_i$ are equivalent to irreducible representations. It will be useful in later sections to know that the (real) irreducible representations of the cyclic group $C_N$ are the rotation matrices with angle $\frac{2\pi}{N}$.

**ForwardVAE**   Higgins et al. [2018] provide the framework for linear disentanglement however intentionally restrained from empirical findings. Caselles-Dupré et al. [2019] presented the first results inducing such representations in VAE latent spaces. Through observing transitions induced by actions in a grid world with $G = C_x \times C_y$ structure, their model successfully learns the rotation matrix representations corresponding to the known irreducible representations of $C_N$.

In implementation, the model stores a learnable parameter matrix (a representation) for each possible action, which is applied when that action is observed. Under the reinforcement learning setting of environment-state-action sequences, they have labelled actions for each observation pair, allowing the action selection at each step. This is suitable for reinforcement problems, however cannot be applied to problems which lack action labelling.

Finally, they offer two theoretical proofs centred around linear disentangled representations. We briefly outline the theorems here: 1) Without interaction with the environment (observing action transitions), you are not guaranteed to learn linear disentangled representations with respect to any given symmetry structure. 2) It is impossible to learn linear disentangled representation spaces $\mathcal{Z}$ of order 2 for the Flatland problem, i.e. learn 1D representations for each component cyclic group.

**Further symmetry structures**   ForwardVAE only explored cyclic symmetry structures $G = C_N \times C_N$, albeit with continuous representations that are expressive enough for $SO(2)$. Subsequent work by Quessard et al. [2020] explored (outside of the VAE framework) linear disentangled representations of $SO(2)$ and non-abelian (non-commutative) $SO(3)$. Similar to ForwardVAE, they required knowledge of the action at each step.

**Learning Observed Actions**   Section 5 revolves around predicting observed actions, a concept which has some prior work. Rybkin et al. [2018] learn a (composable) mapping from pre to post-action latents, conditioned on observing the action. Edwards et al. [2019] learn both a forward dynamics model to predict post action states (given state and action) and a distribution over actions given the initial state. Contrary to our work, both methods allow arbitrarily non-linear actions (parametrised by neural networks) which makes them unsuitable for our task. Furthermore they differ significantly in implementation. Thomas et al. [2017] utilise an autoencoder with a 'disentanglement' objective to encourage actions to correspond to changes in independent latent dimensions. They use a similar reward signal to that we use in Section 5, however have access to labelled actions at each step and encourage no latent structure other than single latents varying with single actions. Choi et al. [2018] learn to predict actions between two frames in an (action) supervised manner with a focus on localising the agents in the image.

## 3   Which Spaces Admit Linear Disentangled Representations

This section empirically explores admission of irreducible representations in latent spaces. In particular we look at standard VAE baselines and search for known cyclic symmetry structure.

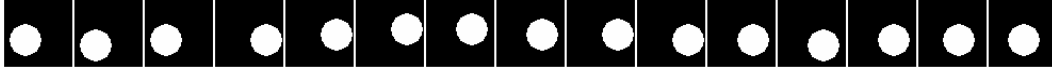

Figure 1: Sequential observations from the Flatland toy problem. If the agent would contact the boundary, it is instead warped to the opposing side (e.g. between the third and fourth observations).

**Problem Setting**    We shall use the Flatland problem [Caselles-Dupré et al., 2018] for consistency with ForwardVAE, a grid world with symmetry structure $G = C_x \times C_y$, manifesting as translation (in 5px steps) of a circle/agent (radius 15px) around a canvas of size $64px \times 64px$. Under the SBDRL framework, we have world space $\mathcal{W} = \{(x_i, y_i)\}$, the set of all possible locations of the agent. The agent is warped to the opposite boundary if within a distance less than $r$. The observation space $\mathcal{O}$, renderings of this space as binary images (see Figure 1), is generated with the PyGame framework [Shinners] which represents the generative process $b$. The inference process $h$ is parametrised by candidate VAE models, specifically the encoder parameters $\theta$. The order of the cyclic groups is given by $(64 - 2 * 15)/5 \approx 7$ which leads to the approximate phase angle of $\alpha \approx 0.924$. All candidate representation spaces shall be restricted to 4 dimensional for simplicity and consistency. All experiments in this and later sections report errors as one standard deviation over 3 runs, using randomly selected validation splits of 10%. We shall evaluate the following baselines, VAE [Kingma and Welling, 2014], $\beta$-VAE [Higgins et al., 2017], CC-VAE [Burgess et al., 2018], FactorVAE [Kim and Mnih, 2018] and DIP-VAE-I/II [Kumar et al., 2017a].

**Evaluation Method**    Once we have a candidate representation (latent) space $\mathcal{Z}$, we need to locate potential irreducible representations $\rho(g)$ of action $a$ by group element $g$. For the defined symmetry structure $G$, we know that the irreducible representations take the form of 2D rotation matrices. We further restrict to observations of actions by either of the cyclic generators $g_x, g_y$ or their inverses $g_x^{-1}, g_y^{-1}$. Consequently, we store 4 learnable matrices to represent each of these possible actions. Since there is no requirement for representations to be admissible on axis aligned planes, we also learn a change of basis matrix for each representation. To locate group representations, we encode the pre-action observation and apply the corresponding matrix, optimising to reconstruct the latent representation of the post-action observation. As we iterate through the data and optimise for best latent reconstruction, the matrices should converge towards irreducible representations if admissible.

If cyclic representations are admissible, then our matrices should learn to estimate the post action latent with low error. However since not all models will encode to the same space, we also compare the relative improvement $||\hat{z}_a - z_a||_{\text{rel}}$, the latent error divided by the expected distance between latent codes. We also introduce a metric to explicitly measure the extent to which actions operate on independent subspaces as required by the definition of a linear disentangled representation. We call this metric the independence score and define it as a function of the latent code $z$ and the latent code after applying action $a$, denoted $z_a$, for actions of $G = G_0 \times \cdots \times G_s$,

$$\text{Independence Score} = 1 - \frac{1}{s!} \sum_{i,\,j \neq i} \max_{a \in G_i, b \in G_j} \left( \left| \left| \frac{z - z_a}{||z - z_a||_2} \cdot \frac{z - z_b}{||z - z_b||_2} \right| \right| \right) \quad . \tag{1}$$

**Results**    For comparison, we first discuss results with ForwardVAE, where cyclic representations are known to reside in axis aligned planes. We provide in supplementary section B explicit errors for post action estimation whilst restricted to axis aligned planes. Comparing ForwardVAE with baseline VAEs, Table 1 reports mean reconstruction errors for the post action observations $x_a$, post action latent codes $z_a$, the estimated phase angle for cyclic representations (against known true value $\alpha$) and the independence score. We can see that none of the standard baselines achieve reconstruction errors close to that of ForwardVAE. Neither do we see good approximations of the true cyclic representation angles ($\alpha$) in any other model. Whilst the mean $\alpha$ learnt by the VAE and $\beta$-VAE are close to the ground truth, the deviation is very large. Dip-II achieves lower error however it is still an order of magnitude worse than ForwardVAE. This is further reflected in the independence scores of each model, with ForwardVAE performing strongly and consistently whilst the baselines perform worse and with high variance. These results suggests that the baseline models do not effectively learn linear disentangled representations.

Table 1: Mean reconstruction values through learnt estimation of a cyclic representation.

| | Forward | VAE | $\beta$-VAE | CC-VAE | FactorVAE | DIP-I | DIP-II |
|---|---|---|---|---|---|---|---|
| $\|\hat{x_a} - x_a\|_1$ | $\mathbf{0.011}_{\pm.004}$ | $0.096_{\pm.004}$ | $0.108_{\pm.020}$ | $0.093_{\pm.015}$ | $0.060_{\pm.027}$ | $0.054_{\pm.011}$ | $0.045_{\pm.013}$ |
| $\|\hat{z_a} - z_a\|_1$ | $\mathbf{0.034}_{\pm.023}$ | $0.328_{\pm.029}$ | $0.255_{\pm.037}$ | $0.151_{\pm.028}$ | $1.286_{\pm.748}$ | $0.431_{\pm.140}$ | $0.270_{\pm.106}$ |
| $\|\hat{z_a} - z_a\|_{rel}$ | $\mathbf{0.054}_{\pm.006}$ | $0.202_{\pm.078}$ | $0.537_{\pm.242}$ | $0.337_{\pm.043}$ | $0.161_{\pm.032}$ | $0.431_{\pm.088}$ | $0.614_{\pm.147}$ |
| $\|\hat{\alpha} - \alpha\|_1$ | $\mathbf{0.009}_{\pm.013}$ | $0.054_{\pm.366}$ | $0.070_{\pm.316}$ | $0.100_{\pm.297}$ | $0.260_{\pm.190}$ | $0.139_{\pm.147}$ | $0.051_{\pm.062}$ |
| Indep | $\mathbf{0.926}_{\pm.063}$ | $0.791_{\pm.109}$ | $0.581_{\pm.475}$ | $0.289_{\pm.465}$ | $0.547_{\pm.362}$ | $0.655_{\pm.216}$ | $0.814_{\pm.119}$ |

Table 2: Disentanglement metrics for baseline models on Flatland and the spearman correlation of independence score with each (italics indicate low confidence).

| | Beta | Mod | SAP | MIG | DCI | FL | Indep |
|---|---|---|---|---|---|---|---|
| Forward | $\mathbf{1.000}_{\pm.001}$ | $\mathbf{0.977}_{\pm.002}$ | $0.301_{\pm.080}$ | $0.021_{\pm.013}$ | $\mathbf{0.960}_{\pm.013}$ | $\mathbf{0.320}_{\pm.003}$ | $\mathbf{0.989}_{\pm.004}$ |
| VAE | $0.876_{\pm.006}$ | $0.387_{\pm.055}$ | $0.296_{\pm.161}$ | $0.044_{\pm.010}$ | $0.010_{\pm.011}$ | $0.697_{\pm.024}$ | $0.625_{\pm.167}$ |
| $\beta$-VAE | $0.954_{\pm.079}$ | $0.698_{\pm.239}$ | $\mathbf{0.620}_{\pm.165}$ | $\mathbf{0.087}_{\pm.110}$ | $0.221_{\pm.165}$ | $0.539_{\pm.239}$ | $0.808_{\pm.250}$ |
| cc-VAE | $0.883_{\pm.198}$ | $0.530_{\pm.356}$ | $0.475_{\pm.387}$ | $0.056_{\pm.062}$ | $0.113_{\pm.159}$ | $0.624_{\pm.211}$ | $0.542_{\pm.186}$ |
| Factor | $0.995_{\pm.005}$ | $0.767_{\pm.099}$ | $0.404_{\pm.056}$ | $0.084_{\pm.093}$ | $0.090_{\pm.024}$ | $0.506_{\pm.130}$ | $0.766_{\pm.079}$ |
| DIP-I | $0.983_{\pm.028}$ | $0.643_{\pm.109}$ | $0.558_{\pm.203}$ | $0.038_{\pm.050}$ | $0.076_{\pm.062}$ | $0.597_{\pm.109}$ | $0.894_{\pm.024}$ |
| DIP-II | $0.795_{\pm.146}$ | $0.292_{\pm.234}$ | $0.252_{\pm.075}$ | $0.057_{\pm.066}$ | $0.044_{\pm.071}$ | $0.762_{\pm.203}$ | $0.760_{\pm.132}$ |
| Corr | $0.743$ | $0.851$ | $0.293$ | $0.180$ | $0.864$ | $-0.855$ | $1.0$ |

# 4 Are Linear Disentangled Representations Advantageous for Classical Disentanglement

This section compares independence and linear disentanglement to a number of classical metrics from the literature. By Locatello et al. [2018], it is known that not all such metrics correlate, and as such it is useful to contrast linear disentanglement with previous understandings.

**Problem Setting** Flatland has two generative factors, the x and y coordinates of the agent. With these we can evaluate standard disentanglement metrics in an effort to discern commonalities between previous understandings. In this work we adapt the open source code of Locatello et al. [2018] to PyTorch and evaluate using the following metrics: Higgins metric (Beta) [Higgins et al., 2017], Mutual Information Gap (MIG) [Chen et al., 2018], DCI Disentanglement metric [Eastwood and Williams, 2018], Modularity (Mod) metric [Ridgeway and Mozer, 2018], SAP metric [Kumar et al., 2017b]. In addition, we evaluate two further metrics that expand on previous works: Factor Leakage (FL) and the previously introduced Independence score.

**Factor Leakage** Our Factor Leakage (FL) metric is descended from the MIG metric which measures, for each generative factor, the difference in information content between the first and second most informative latent dimensions. Considering linear disentangled group representations are frequently of dimension 2 or higher, this would result in low MIG scores, implying entanglement despite being linearly disentangled. We extend it by measuring the information of all latent dimensions for each generative factor (i.e. each group action) and reporting the expected area under this curve for each action/factor. Intuitively, entangled representations encode actions over all dimensions; the fewer dimensions the action is encoded in, the more disentangled it is considered with respect to this metric.

**Results** Table 2 reports classical disentanglement metrics alongside our FL and independence metrics with comparisons of ForwardVAE to baselines. ForwardVAE clearly results in stronger scores with very low deviations under each metric except the SAP and MIG score, both of which measure the extent factors are encoded to single latent dimensions. Naturally, this is not suited to the 2 dimensional representations we expect in Flatland. Indeed, by Caselles-Dupré et al. [2019], no action can be learnt in a single dimension on this problem. We also report correlation of the independence scores against all other metrics. We see strong correlations with all metrics other than

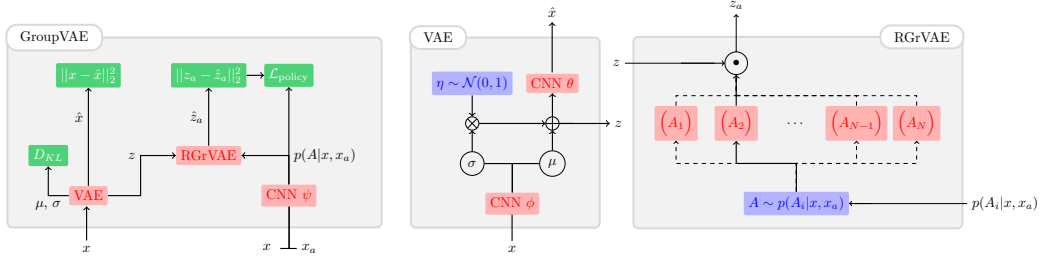

Figure 2: Schematic diagram for RGroupVAE and components. • denotes matrix vector multiplication. Dashes denote possible paths dependent on selected action.

🟥 - Learnable module.     🟩 - Loss.     🟪 - Operation without parameters

SAP and MIG. From the performance of ForwardVAE and the correlation of independence and other metrics, we can see linear disentangled representations are beneficial for classical disentanglement metrics, and appears to result in very low variance metric scores.

## 5   Unsupervised Action Estimation

ForwardVAE explicitly requires the action at each step in order to induce linear disentangled structures in latent spaces. We now show that policy gradients allow jointly learning to estimate the observed action alongside the latent representation mapping. We will then examine properties of the model such as learning over longer term action sequences and temporal consistency. All experimental details are reported in supplementary material section A and code is available at `https://github.com/MattPainter01/UnsupervisedActionEstimation`. Before introducing the model, we briefly outline the policy gradient method that it will utilise.

**Policy Gradients**   Policy gradient methods will allow us to optimise through a Categorical sampling of $\psi$-parametrised distribution $p(A|\psi, s)$ over possible choices $\{A_1, \ldots, A_N\}$ and conditioned on state $s$. The policy gradient loss in the REINFORCE [Williams, 1992] setting where action $A_i$ receives reward $R(A_i, s)$ is given by,

$$\mathcal{L}_{policy} = \begin{cases} -\log(p(A_i|\psi, s)) \cdot R(A_i, s) & \text{if } R(A_i, s) > 0 \\ -\log(1 - p(A_i|\psi, s)) \cdot |R(A_i)| & \text{if } R(A_i, s) < 0 \end{cases} \quad . \tag{2}$$

We find that minimising the regret $R(A_i, s) - \max_j R(A_j, s)$ instead of maximising reward provides more stable learning for FlatLand, however the increased runtime was unacceptable for the more varied dSprites data, which uses standard reward maximisation. To encourage exploration we use an $\epsilon$-greedy policy or subtract the weighted entropy $0.01H(p(A_i|\psi, s))$, a common technique used by methods such as Soft Actor-Critic [Haarnoja et al., 2018].

**Reinforced GroupVAE**   Reinforced GroupVAE (RGrVAE) is our proposed method for learning linear disentangled representation in a VAE without the constraint of action supervision. A schematic diagram is given in Figure 2. Alongside a standard VAE we use a small CNN (parameters $\psi$) which infers from each image pair a distribution over a set of possible (learnable) internal representation matrices $A_i$. These matrices can be restricted to purely cyclic representations by learning solely a cyclic angle, or they can be learnt as generic matrices. We also introduce a decay loss on each representation towards the identity representation, since we prefer to use the minimum number of representations possible. The policy selection distribution is sampled categorically and the chosen representation matrix is applied to the latent code of the pre-action image with the aim of reconstructing that of the post-action image. The policy selection network is optimised through policy gradients with rewards provided as a function of the pre-action latent code $z$, the post action latent code $z_a$ and the predicted post action code $\hat{z}_a$,

$$R(a, x, x_a) = ||z - z_a||_2^2 - ||\hat{z}_a - z_a||_2^2 \quad . \tag{3}$$

We then train a standard VAE with the additional policy gradient loss and a weighted prediction loss given by

$$\mathcal{L}_{pred} = ||\hat{z}_a - z_a||_2^2 \quad , \quad \mathcal{L}_{total} = \mathcal{L}_{VAE} + \mathcal{L}_{policy} + \gamma \mathcal{L}_{pred} \quad . \tag{4}$$

Table 3: Reconstruction errors for post action observation ($x$), latent ($z$) and representation $\alpha$.

(a) FlatLand

|  | ForwardVAE | RGrVAE |
|---|---|---|
| $\|\hat{x}_a - x_a\|_1$ | $\mathbf{0.011}_{\pm 0.004}$ | $0.016_{\pm0.004}$ |
| $\|\hat{z}_a - z_a\|_1$ | $\mathbf{0.034}_{\pm 0.023}$ | $0.100_{\pm0.029}$ |
| $\|\hat{z}_a - z_a\|_{\text{rel}}$ | $\mathbf{0.054}_{\pm0.005}$ | $0.183_{\pm0.034}$ |
| $\|\hat{\alpha} - \alpha\|_1$ | $\mathbf{0.009}_{\pm 0.013}$ | $0.012_{\pm0.040}$ |
| Independence | $0.989\pm0.004$ | $0.960\pm0.015$ |

(b) dSprites

|  | RGrVAE | VAE |
|---|---|---|
| $\|\hat{x}_a - x_a\|_1$ | $\mathbf{0.008}_{\pm0.005}$ | $0.010_{\pm0.000}$ |
| $\|\hat{z}_a - z_a\|_1$ | $\mathbf{0.103}_{\pm0.040}$ | $0.294_{\pm0.106}$ |
| $\|\hat{z}_a - z_a\|_{\text{rel}}$ | $\mathbf{0.407}_{\pm0.100}$ | $0.925_{\pm0.290}$ |
| $\|\hat{\alpha} - \alpha\|_1$ | $\mathbf{0.205}_{\pm0.095}$ | $0.312_{\pm0.159}$ |
| Independence | $0.985_{\pm0.014}$ | $0.879_{\pm0.050}$ |

Table 4: Disentanglement Metrics for RGrVAE. We see similar scores on dSprites and FlatLand, suggesting linear disentanglement in both cases.

|  | Beta | Mod | SAP | MIG | DCI | FL |
|---|---|---|---|---|---|---|
| FlatLand | $1.000_{\pm0.000}$ | $0.956_{\pm0.010}$ | $0.464_{\pm0.059}$ | $0.051_{\pm0.020}$ | $0.809_{\pm0.059}$ | $0.355_{\pm0.016}$ |
| dSprites | $0.998_{\pm0.001}$ | $0.901_{\pm0.010}$ | $0.420_{\pm0.040}$ | $0.033_{\pm0.014}$ | $0.689_{\pm0.057}$ | $0.165_{\pm0.030}$ |

**Flatland** To demonstrate the effectiveness of the policy gradient network, we evaluate RGrVAE on Flatland. We allow 4 latent dimensions and initialise 2 cyclic representations per latent pair with random angles but alternating signs to speed up convergence. Examples of the learnt actions are given in supplementary section C. Table 3a reports reconstruction errors showing RGrVAE achieves similar error rates to ForwardVAE. Whilst latent reconstructions are similar to some baselines, it is much more independent in these (and all) cases. This proves the basic premise that RGrVAE can induce linear disentangled representations without action supervision.

**dSprites** Table 3b reports reconstruction errors on dSprites, with symmetries in translation, scale and rotation leading to symmetry structure $G = C_3 \times C_{10} \times C_8 \times C_8$. We see that the symmetry reconstruction is an order of magnitude worse than on FlatLand which is expected since it is a harder problem. Whilst we see similar latent/relative reconstruction as on FlatLand, the observation reconstruction appears much closer to the baseline. This is due to the object taking up a significantly smaller fraction of the image in dSprites than in FlatLand, so the expected distance between pre and post action observations is much lower. We do note that RGrVAE continues to have highly independent representations compared to the baseline. This combined with action traversals in Figure 3 (see supplementary material C for full traversals) and disentanglement metrics (next paragraph) demonstrate that RGrVAE has extracted 4 symmetries, the most currently demonstrated directly from pixels. Note from the traversals, however that the rotation symmetry learnt was $C_4$, due to spatial symmetries of the square and oval resulting in degenerate actions and their structures being $C_4/C_5$, not $C_{10}$. It is possible with additional training and learning rate annealing this could be overcome, but likely with significantly increased runtime. We believe a comparably trained ForwardVAE would perform similarly or better thus our comparison to a baseline in this experiment.

**Classical Disentanglement Metrics** Table 4 reports disentanglement metrics for RGrVAE on FlatLand and dSprites. On FlatLand, we can see that despite removing the requirement of action supervision, RGrVAE performs similarly to ForwardVAE (Table 2) on the Beta, Modularity, FL and Independence metrics. The SAP score is more comparable to baseline methods, implying that despite the high independence and low cyclic reconstruction errors, RGrVAE may rely more on single dimensions than ForwardVAE. The DCI disentanglement is also lower and the MIG larger, which are both consistent with this. Despite this, the DCI disentanglement is comparatively much larger than baselines, which shows, with the Beta, FL and independence scores that RGrVAE does not encode factors in solely single dimensions, instead capturing a linear disentangled representation. We also include metrics evaluated on dSprites where we see similar scores to FlatLand and a similar level of consistency in the scores which was not as evident in baselines. This provides further evidence that RGrVAE learns linear disentangled representations effectively.

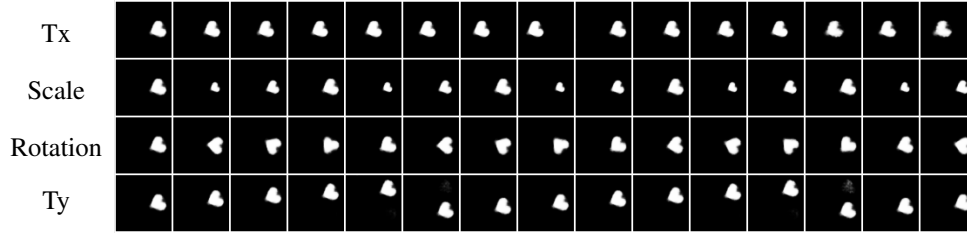

Figure 3: Action traversals for main actions of RGrVAE trained on dSprites. We sampled the true actions from the dataset based on the following symmetry groups: Scale: $C_3$, Rotation: $C_{10}$, Translation: $C_8$.

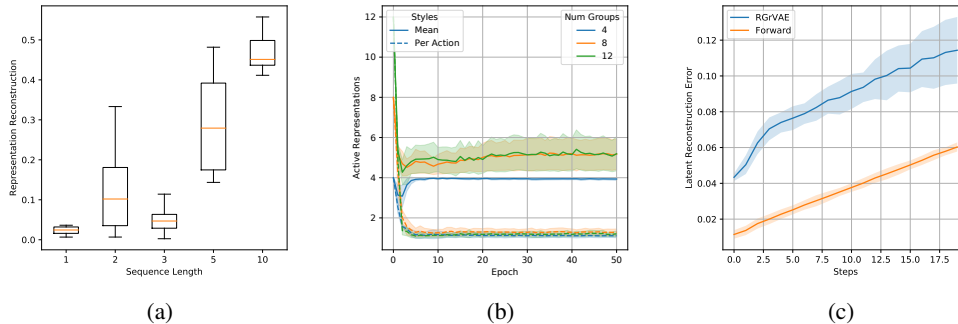

Figure 4: Properties of RGrVAE. a) $\alpha$ reconstruction MSE vs sequence length. b) Estimated active number of representations in total (solid) and per action (dashed) over training for different total available representations. c) Latent reconstruction MSE for applying actions over a number of steps.

**Longer Action Sequences** By applying a chosen action and decoding the result, we have a new image pair (with the target) from which we can infer another action. Repeating this allows us to explore initial observation pairs which differ by more than the application of a single action. In the limit this could remove the requirement of having sequential data, however by [Caselles-Dupré et al., 2019], we know that this no longer guarantees linear disentanglement with respect to any particular symmetry structure. Figure 4a reports the $C_N$ $\alpha$ reconstruction error, where we see that larger steps results in gradually degraded estimation, however it is relatively consistent for a small number of steps. Note that this does not preclude linear disentanglement, simply linear disentanglement with respect to $G = C_x \times C_y$ is extremely unlikely. Since our measure is specific to $G$, linear disentanglement with respect to other symmetry structures would not be evident from this figure.

**Over Representation** Thus far we have not been concerned with the choice of how many internal representations to allow RGrVAE. In many cases we will not know the exact symmetry structure so a 1 to 1 mapping of internal representations to expected actions on the data will not be possible. We now explore over-representation, the case where the number of internal representations is greater than the number of expected true actions. In Figure 4b we plot an estimate of the number of total/per-action active representations ($N \approx e^h$ where $h$ is the mean or per action entropy). Over training we see the number of active representations decrease, towards 1 for each action but between 4 and 6 for the total. This total is higher than the expected 4. Since for each action there is close to 1 active representation we believe it is increased by policies for different actions choosing the same minority representation some small fraction of the time. These findings show that as long as we choose a number of internal representations greater than the number of true actions then we can still learn.

**Temporal Consistency** A desirable property of representing actions is that they remain accurate over time - the representation isn't degraded as we apply new actions. Figure 4c reports the latent reconstruction error $||\hat{z}_a - z_a||_1$ as additional actions are applied. We can see that the quality of representations degrades gradually as we add steps and indeed is only marginally worse than the error achieved by ForwardVAE. Note that for this test, only the initial observation is provided to the model,

Table 5: Performance of RGrVAE under visual noises. $\tau$ is the number of epochs to 0.95 estimated independence, representing policy network convergence.

|  | None | Gaussian | Salt | Backgrounds |
|---|---|---|---|---|
| True Indep | $0.9595_{\pm 0.005}$ | $0.9325_{\pm 0.020}$ | $0.9329_{\pm 0.011}$ | $0.9071_{\pm 0.055}$ |
| $\|\|\hat{\alpha} - \alpha\|\|_1$ | $0.0315_{\pm 0.021}$ | $0.0278_{\pm 0.010}$ | $0.0174_{\pm 0.021}$ | $0.0312_{\pm 0.017}$ |
| $\tau_{0.95}$ | $176.0_{\pm 80.2}$ | $164.33_{\pm 50.2}$ | $168.67_{\pm 38.6}$ | $919.0_{\pm 644.2}$ |

unlike our experiment with longer action sequences, where predicted observations are fed back into it as it works towards a terminal observation.

**Convergence Consistency and Robustness** It is important to understand robustness of our models and one way to do that is to measure performance under less than ideal conditions. We will introduce different forms of visual noise to the FlatLand problem and find the conditions under which RGrVAE does and doesn't converge. We will first consider simple Gaussian and Salt+Pepper noises before looking at adding complex distractors through real world backgrounds. Note that for these tests we slightly increased the complexity of the underlying VAE by doubling the channels (to 64 from 32) for the intermediate/hidden layers (i.e. not output or input). This was since we assumed that more complex problems would converge faster with (slightly) more complex models.

We find in Table 5 that the addition of simple noise (salt and pepper, Gaussian) did not prevent policy convergence and results in strong independence and reconstruction scores. Whilst the independence score was reduced slightly by all noise types, the symmetry reconstruction remains similar to the noiseless case. We further report the mean number of epochs to reach an estimated independence of 0.95 ($\tau_{0.95}$), which represents the convergence of the policy network. Here we can see that the simple noises did not change the convergence rate but the complex distractors (backgrounds) resulted in a order of magnitude slower convergence. We present additional data in supplementary section B.

On consistency, we would like to highlight that we found low learning rates for the VAE and policy network (lr $\approx 10^{-4}$) with high learning rate for the internal RGrVAE matrix representations (lr $\approx 10^{-2}$) to be very beneficial for consistent convergence (regardless of task) and generally prevented convergence to suboptimal minima. When the learning rates are equal we observed convergence to suboptimal minima every few runs.

## 6 Conclusion

Symmetry based disentangled representation learning offers a framework through which to study linear disentangled representations. Reflected in classical disentanglement metrics, we have shown empirically that such representations are beneficial and very consistent. However, we find that even for simple problems, linear disentangled representations are not present in classical VAE baseline models, they require structuring the loss to favour them. Previous works achieved this through action supervision and imposing a reconstruction loss between post action latents and their predicted values after applying a learnt representation to the pre action latents. We introduced our independence metric to measure the quality of linear disentangled representations, before introducing Reinforce GroupVAE to induce these representations without explicit knowledge of actions at every step. We show that RGrVAE prefers to learn internal representations that reflect the true symmetry structure and ignore superfluous ones. We also find that it still performs when observations are no longer separated by just a single action, and can model short term action sequences, however with decreasing ability to recover the chosen symmetry due to the reduced bias. We demonstrate its ability to learn up to 4 symmetries directly from pixels and show that it is robust under noise and complex distractors.

Extracting linear disentangled representations is achievable when the underlying symmetry groups can be sampled individually or in small groups. Whilst this is likely possible for video data since over short time scales we might expect only a few actions to occur simultaneously, this may not always be the case. We believe future work should focus on extracting the simplest symmetry structure that is sufficient to explain the data, ideally without prior definition of what this structure should be. The ability to infer observed actions may be vital in such a system, and we view our work as moving towards this goal.

# 7 Acknowledgements

Funding in direct support of this work: EPSRC Doctoral Training Partnership Grant EP/R512187/1. The authors also acknowledge the use of the IRIDIS High Performance Computing Facility, and associated support services at the University of Southampton, in the completion of this work.

# 8 Broader Impact

Representation learning as a whole does have the potential for unethical applications. Disentanglement if successfully applied to multi-object scenes could allow (or perhaps require, this is unclear) segmentation/separation of individual objects and their visual characteristics. Both segmentation and learning visual characteristics have numerous ethical and unethical uses on which we wont speculate. For our particular work, we don't believe understanding and encouraging linear disentangled representations has any ethical considerations beyond the broad considerations of disentanglement work as a whole. We do believe that routes towards reducing the degree of human annotation in data (such as our proposed model) is beneficial for reducing human bias, although this can introduced even by the choice of base data to train on. Unfortunately for our work (and most unsupervised work), explicit supervision allows for more rapid convergence and consequently a lower environmental impact, a topic which is of increasing concern especially for deep learning which leans heavily on power intensive compute hardware.

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
