[Supplementary Material]

# Supplementary Material to Linear Disentangled Representations and Unsupervised Action Estimation

**Matthew Painter, Jonathon Hare, Adam Prügel-Bennett**
Department of Electronics and Computer Science
Southampton University
{mp2u16, jsh2, apb}@ecs.soton.ac.uk

## A    Experimental Details

This section lists the experimental details for each figure in the main paper alongside addition information.

**Tables 1, 2, 3a, 4**    All FlatLand results were run with the following parameters: learning rate 1e-4, 100 epochs, 4 latent dimensions, Adam optimiser, 3 repeats, MSE loss. Estimations $\hat{\alpha}$ were trained for 5 epochs using the Adam optimiser with MSE loss and learning rate 0.1. RGrVAE used a latent reconstruction weight $\gamma = 4$.

Table A.1: $\beta$ choices for each VAE model. [†] Capacity 15, 3500 step leadin

|          | Forward | RGrVAE | VAE | $\beta$-VAE | CC-VAE | FactorVAE | DIP-I | DIP-II |
|----------|---------|--------|-----|-------------|--------|-----------|-------|--------|
| $\beta$  | 1       | 1      | 1   | 5           | 1000[†] | 1         | 1     | 1      |

**Tables 3b, 4. Figure 3**    All dSprites experiments were ran with the following parameters: learning rate 1e-4, 16 latent dimensions, 5M iterations, Adam optimiser, 3 repeats, bce loss.

**Figure 4**    All results use the same setup as the previous tables with the following exception: Figure 3a, RGrVAE group structure is two cyclic representations alongside an identity representation, for each latent pair.

**Table 5**    All results use the same setup as previous FlatLand experiments except for runtime. All noise types except real world backgrounds were ran for 3K iterations. Real world backgrounds were run for 20K iterations.

**Model Architectures**    We now detail specific architectures used through our experiments. Where possible we have used equivalent backbone architectures or author reference implementations. We list each architecture:

- ForwardVAE [2]: Author implementation found `https://github.com/Caselles/NeurIPS19-SBDRL` and detailed in Table A.2.
- RGrVAE: Reuses the backbone architecture of ForwardVAE alongside the action encoder given in Table A.3.
- Dip-VAE [6]: Reuses the backbone architecture of ForwardVAE.
- FactorVAE [4]: Reuses the backbone architecture of ForwardVAE alongside discriminator given in Table A.4.

- cc-/$\beta$-/VAE [1, 3, 5]: Implementation by Higgins et al. [2017] and seen in Table A.5

Table A.2: Underlying architecture of ForwardVAE. All layers other than final encoder and decoder layers use a SELU non linearity. Intermediate layers were modified for the noise experiments as detailed in the main paper.

| ForwardVAE | | | | | |
|---|---|---|---|---|---|
| Encoder | | | Decoder | | |
| Conv | 32 | stride 2, kernel size 4 | Linear | 256 | - |
| Conv | 32 | stride 2, kernel size 4 | Linear | 512 | - |
| Conv | 32 | stride 2, kernel size 4 | ConvT | 32 | stride 2, kernel size 4 |
| Conv | 32 | stride 2, kernel size 4 | ConvT | 32 | stride 2, kernel size 4 |
| Linear | 256 | - | ConvT | 32 | stride 2, kernel size 4 |
| Linear | 256 | - | ConvT | 1 | stride 2, kernel size 4 |
| Linear | $2N_{\text{latents}}$ | - | | | |

Table A.3: Action encoder ($\psi$) for RGrVAE . All layers other than final use a ReLU non linearity.

| RGrVAE - Action Encoder | | |
|---|---|---|
| Conv | 32 | stride 2, kernel size 3 |
| Conv | 16 | Stride 2, Kernel Size 3 |
| Conv | 16 | Stride 2, Kernel Size 3 |
| Linear | $N_{\text{actions}}$ | - |

Table A.4: FactorVAE discriminator . All layers other than final use a LeakyReLU (0.2) non linearity.

| FactorVAE - Discriminator | | |
|---|---|---|
| Linear | 1000 | - |
| Linear | 1000 | - |
| Linear | 1000 | - |
| Linear | 1000 | - |
| Linear | 1000 | - |
| Linear | 2 | - |
| Softmax | - | - |

Table A.5: Underlying architecture of cc-/$\beta$-/VAE. All layers other than final encoder and decoder layers use a ReLU non linearity.

| cc-/$\beta$-/VAE | | | | | |
|---|---|---|---|---|---|
| Encoder | | | Decoder | | |
| Conv | 32 | stride 2, kernel size 4 | Linear | 256 | - |
| Conv | 32 | stride 2, kernel size 4 | Linear | 1024 | - |
| Conv | 64 | stride 2, kernel size 4 | ConvT | 64 | stride 2, kernel size 4 |
| Conv | 64 | stride 2, kernel size 4 | ConvT | 32 | stride 2, kernel size 4 |
| Linear | 256 | - | ConvT | 32 | stride 2, kernel size 4 |
| Linear | $2N_{\text{latents}}$ | - | ConvT | 1 | stride 2, kernel size 4 |

# B Additional Experiments

**Axis aligned alpha reconstruction** The table below reports errors for reconstructing post action latents whilst restricted to purely axis aligned planes.

Table B.1: Validation reconstruction MSE between predicted z (after applying cyclic representation) and true z (encoding post-action image) in each latent plane, alongside the expected distance between any two latent codes over the dataset. Standard VAEs never achieve low errors, cyclic representations are not present. Actions $a$: 0 - up, 1 - left, 2 - down, 3 - right.

| | VAE | | | | | | ForwardVAE | | | | | |
|---|---|---|---|---|---|---|---|---|---|---|---|---|
| a | 0, 1 | 0, 2 | 0, 3 | 1, 2 | 1, 3 | 2, 3 | 0, 1 | 0, 2 | 0, 3 | 1, 2 | 1, 3 | 2, 3 |
| 0 | 0.711 | 0.718 | 0.714 | 0.718 | **0.524** | 0.718 | **0.006** | 0.525 | 0.525 | 0.529 | 0.525 | 0.525 |
| 1 | **0.755** | 0.913 | 0.898 | 0.918 | 0.912 | 0.918 | 0.475 | 0.475 | 0.478 | 0.474 | 0.476 | **0.004** |
| 2 | 0.770 | 0.786 | 0.770 | 0.784 | **0.609** | 0.785 | **0.006** | 0.529 | 0.529 | 0.531 | 0.528 | 0.529 |
| 3 | **0.678** | 0.878 | 0.871 | 0.881 | 0.875 | 0.881 | 0.476 | 0.475 | 0.487 | 0.479 | 0.476 | **0.004** |
| | Independence: 0.791 | | | | | | Independence: 0.926 | | | | | |
| | Expected distance: 0.821 | | | | | | Expected distance: 0.502 | | | | | |

**Different RGrVAE Representations** We present a brief exploration allowing more expressivity in RGrVAE internal representations. In the main paper, these representations were solely cyclic, where the phase angle $\alpha$ is the only learnable parameter. We now explore generic matrices such as those used by ForwardVAE. We report in Table B.2 the disentanglement scores for the cyclic representations verses the generic matrix representations. Both methods achieve similar results, the major difference between them is in convergence rate. Figure B.1 compares an estimated independence score over training for each method. The cyclic representations converge extremely quickly, whereas the matrix representations get stuck in local minima for long periods of time before eventually converging to the global minima.

Figure B.1: Estimated independence score over training for different internal RGrVAE representation structures. 'c' denotes standard cyclic representation. 'c+/-' denotes initialised cyclic representation to positive/negative. 'm' denotes generic 2D matrix representation. 'ddn' denotes (not learnable) representation of reflection.

**Attention** Attentional mechanisms offer alternate means to allow gradient through a distribution over choices. Instead of sampling the distribution and using policy gradients, attention forms a linear combination of the choices weighted by the distribution. In our case we are interested in

Table B.2: Disentanglement metric scores for the different internal representation choices.

| Metric | Cyclic | Cyclic Initialised | Matrices | Cyclic + Reflection |
|---|---|---|---|---|
| Beta | $1.000_{\pm.000}$ | $1.000_{\pm.000}$ | $1.000_{\pm.000}$ | $1.000_{\pm.000}$ |
| MIG | $0.096_{\pm0.030}$ | $0.055_{\pm0.028}$ | $0.049_{\pm0.030}$ | $0.085_{\pm0.020}$ |
| DCI | $0.814_{\pm0.077}$ | $0.747_{\pm0.070}$ | $0.738_{\pm0.056}$ | $0.770_{\pm0.108}$ |
| Mod | $0.954_{\pm0.004}$ | $0.953_{\pm0.015}$ | $0.950_{\pm0.012}$ | $0.952_{\pm0.017}$ |
| SAP | $0.500_{\pm0.043}$ | $0.559_{\pm0.019}$ | $0.584_{\pm0.012}$ | $0.540_{\pm0.160}$ |
| FL | $0.346_{\pm0.025}$ | $0.369_{\pm0.010}$ | $0.356_{\pm0.025}$ | $0.346_{\pm0.016}$ |
| Indep | $0.905_{\pm0.068}$ | $0.932_{\pm0.029}$ | $0.908_{\pm0.045}$ | $0.924_{\pm0.061}$ |

(a) Gaussian      (b) Salt and Pepper      (c) Backgrounds

Figure B.2: Noise types

learning the correct irreducible representations which requires each representation to learn exactly the correct cyclic phase angle. When we predict post-action latent codes through a linear combination of representations, we lose the guarantee that the gradient will point towards this solution. Since reinforce applies solely one representation exactly once, we are guaranteed that (if the policy is accurate and the latent structure is amenable) the gradient will point towards this solution. We find that the cyclic representation error $||\hat{\alpha} - \alpha|| = 0.157$ is far worse than the $0.012$ error of RGrVAE. Furthermore, the independence score is $0.830_{\pm0.109}$ which is comparatively low compared to RGrVAE ($0.955_{\pm0.014}$) which larger deviation. These statistics showed us that the reinforcement method was a better candidate to learn linear disentangled representations.

**Visual Noise** It is important to understand robustness of our models and one way to do that is to measure performance under less than ideal conditions. We will introduce different methods of visual noise to the FlatLand problem and find the conditions under which RGrVAE does and doesn't converge. We will first consider simple Gaussian and Salt+Pepper noises before looking at adding complex distractors through real world backgrounds. Note that for these tests we slightly increased the complexity of the underlying VAE by doubling the channels (to 64 from 32) for the intermediate/hidden layers (i.e. not output or input). This was since we assumed that more complex problems would converge faster with (slightly) more complex models. We also note that batch size for this experiment was 1024 compared to 128 used for most other experiments. Whilst this doesn't seem to effect performance it does mean that the $\tau$ scores shouldn't be directly compared to epochs in other figures without appropriate scaling.

We find that simple noises (Gaussian/Salt and Pepper) do little to hinder policy network convergence (est. indep) and results in strong independence and reconstruction scores. We also report $\tau_{0.95}/\tau_{0.90}$, the average number of epochs to 0.95/0.90 estimated independence which represents convergence speed of the policy. Again, the simple noises did not reduce the speed of policy convergence either.

Table B.3: Performance of RGrVAE under visual noises. All results are taken across 3 runs and 300 epochs.

| | None | Gaussian | Salt | Backgrounds |
|---|---|---|---|---|
| Est. Indep | $0.9877_{\pm0.006}$ | $0.9916_{\pm0.004}$ | $0.9908_{\pm0.005}$ | $0.9922_{\pm0.002}$ |
| True Indep | $0.9595_{\pm0.005}$ | $0.9325_{\pm0.020}$ | $0.9329_{\pm0.011}$ | $0.9071_{\pm0.055}$ |
| $||\hat{x}_2 - x_2||_1$ | $0.0186_{\pm0.004}$ | $0.0168_{\pm0.001}$ | $0.087_{\pm0.003}$ | $0.0073_{\pm0.001}$ |
| $||\hat{z}_2 - z_2||_1$ | $0.1951_{\pm0.03}$ | $0.1560_{\pm0.008}$ | $0.1270_{\pm0.009}$ | $0.0510_{\pm0.006}$ |
| $||\hat{\alpha} - \alpha||_1$ | $0.0315_{\pm0.021}$ | $0.0278_{\pm0.010}$ | $0.0174_{\pm0.021}$ | $0.0312_{\pm0.017}$ |
| $\tau_{0.95}$ | $176.0_{\pm80.2}$ | $164.33_{\pm50.2}$ | $168.67_{\pm38.6}$ | $919.0_{\pm644.2}$ |
| $\tau_{0.90}$ | $171.33_{\pm78.0}$ | $141.67_{\pm46.9}$ | $163.67_{\pm36.1}$ | $734.0_{\pm601.0}$ |

Upon introducing backgrounds convergence became (unsurprisingly) harder to achieve. On top of the increased complexity we further added 2 latents and doubled channels in the linear and convolution layers in the decoder since rapidly learning to reconstruct images allows the MSE reconstruction signal to the matrix representations to get stronger in relation. Obviously the convergence times increased by an order of magnitude, we believe due to the matrix representation gradient being small compared to the usual. When the representations converge slowly then the policy converges slowly and the incentive to arrange the latent space correctly decreases. This all results in generally very slow convergence. Despite the slow convergence, this type of noise reduced performance on the true independence but not by an extreme amount and indeed scored better on observation reconstruction likely since this was dominated by reconstructing the background.

**Additional symmetries** In Figure C.2 we provide action traversals for RGrVAE on dSprites. We should note that we increased the step size for translations, scales and rotations so that the rotations are of higher degree. This was for demonstration purposes and we believe the model would still learn when varying only a single index in the generative factors at a time. We also removed the change of basis for this experiment, which is why the actions are learnt in neighbouring dimensions to their inverses.

# C Action Traversals

Figure C.1: Actions learnt by RGrVAE corresponding with the environment actions up, down, left and right. Note the wrapping at boundaries which is expected by the symmetry structure.

Figure C.2: Action traversals for all available actions of RGrVAE trained on dSprites and a heatmap over the actions selected over the dataset. We sampled the true actions from the dataset based on the following symmetry groups: Scale: $C_3$, Rotation: $C_{10}$, Translation: $C_8$. Thus the group acting on the data is: $G = C_3 \times C_{10} \times C_8 \times C_8$.