[Reviews · NeurIPS 2020]

Review 1

Summary and Contributions: This paper analyses new and existing VAE methods, finding that many do not naturally find irreducible representations, but that they can be improved by changing the loss landscape. This paper also proposes a new disentanglement metric that measures the independence of group actions on the learned representation. Finally the paper proposes a method to recover actions from transitions when no labelling is available, assuming the actions correspond to independent observational components of the environment. Rebuttal: Thank you for your response, I continue to support acceptance for this paper.

Strengths: This paper does an interesting follow up and analysis on existing methods, and then provide an interesting addition. The analysis itself is insightful; taking a very simple setting, how do various VAE/disentangling methods succeed and fail? The results are not extremely surprising but they provide more evidence, which is good. The novel algorithmic contributions of this paper are also interesting, in the sense that they add to the space of VAE. Although attempting to recover actions from observational data is not novel from an RL perspective, within the space of disentanglement literature this is a neat contribution. Another strength of this paper is that I found it to be very well written, clear & concise.

Weaknesses: It's not clear to me whether we're starting to overfit to such simple disentanglement cases as a blob on a 2d grid (well, in a sense we already have, see e.g. [4]) where we recover x & y, or whether it's valuable to pursue those simplest cases to the limit where we understand them and where our methods should pass these like sanity checks. I say this because while I see the value of such setups where we control and understand everything, the problems we care about in the end are not that clean, so is it worth pursuing the simple settings to the limit? Another weakness-that's-not-necessarily-a-weakness is that the algorithmic contributions of this paper borrow a lot from existing ideas, and are thus of limited novelty. Related to this is the lack of comparisons with such existing ideas. While the authors do a good job of referring to the disentanglement literature, parallels to the RL literature are lacking (see below).

Correctness: I could not find anything wrong.

Clarity: The paper is very well written and clear.

Relation to Prior Work: The second part of this work which aims at recovering the action space of an interactive agent is reminiscent of several prior works [1-5]. Although in this work the actions taken are unknown, the rewards used to recover which actions were taken is similar to ones use in some of these works [1,3,4] to reward disentangled feature/policy pairs. It may be interesting to compare to/consider them, especially considering that the proposed method seems to have the exact same weaknesses; an almost perfect disentanglement in simple environments such as gridworlds, the occasional suboptimal minima where learning gets stuck with redundant or mis-disentangled actions/policies, and an inability to deal with longer action sequences correctly. An interesting, if unsatisfactory conclusion from these works is that such approaches do not cleanly scale to more complex observation and action spaces. I wonder if the same is true here. Maybe less relevant, but I am also reminded of Independent Mechanisms [6]. [1] Learning what you can do before doing anything, Rybkin et al, ICLR 2019 [2] Contingency-aware exploration in reinforcement learning, Choi et al, ICLR 2019 [3] Disentangling Controllable and Uncontrollable Factors by Interacting with the World, Sawada et al, NeurIPS 2018 [4] Independently controllable factors, Thomas et al., 2017 [5] Imitating Latent Policies from Observation, Edwards et al, ICML 2019 [6] Learning Independent Causal Mechanisms, Parascandolo et al, ICML 2018

Reproducibility: Yes

Additional Feedback: I do think it would be valuable to understand how brittle the proposed method is, or to conjecture more on its weaknesses. I don't think experiments on something like Atari are necessary nor would be valuable, but adding a bit of noise, confounders, or observational complexity would probably be quite informative as to the robustness of the various methods.


Review 2

Summary and Contributions: The paper investigates how the concept of Symmetry Based Disentangled Representation Learning (SBDRL) relates to other popular disentanglement metrics and representation learning methods. In particular it demonstrates that 1) models such as the VAE do not learn linear disentangled representations, unless explicitly biased to, and 2) linear disentangled representations seem to improve already established metrics such as the mutual information gap (MIG). The authors build up on an existing Forward VAE model and describe an unsupervised method to train it.

Strengths: The paper clearly demonstrates some intuitive, but not explicitly shown so far results, regarding linear disentangled features. The authors provide substantial empirical evidence demonstrating that learning linear disentangled features requires explicit inductive bias. It also compares SBDRL with other popular methods using multiple disentanglement metrics.

Weaknesses: The paper presents RGrVAE which is essentially the same model as Forward VAE where the action is also treated as a latent variable. Given the discrete nature of the action space it is quite a natural step to model that latent variable through a categorical distribution and infer the parameters of that distribution through REINFORCE in an end-to-end fashion. However, one of the key questions in this approach is how do you choose the number of actions N (i.e. number of clusters?). Despite that the paper does not mention anything about that decision. Perhaps utilising a Dirichlet process might be beneficial. My biggest concern though is that the authors apply the proposed unsupervised method only to a single task which is the task introduced in the Forward VAE paper. Training with REINFORCE is known to be quite unstable so I definitely would like to see the behaviour of the algorithm with at least one other tasks. Perhaps considering the dSPRITE dataset can be another task to look into?

Correctness: Tthe empirical methodology adequately address the core hypotheses of the paper. Overall, I do not find any major issues with the correctness of the paper.

Clarity: The paper is written well, however, there are some places where a few things need to be clarified. For example: - Why is the order of the group 7? Where does this number 5 that you dived by come from? (line 140) - What does exactly the dot operator represent within the max operator of equation (2)? - I am not sure I can follow the reasoning in line 196. Why isn't MIG expressive enough? - Why is the reward proportional to the L2 norm of the difference between the previous and current latent codes? Don't you want the predicted post action code (based on applying the learnt group operator and the previous latent code) to be as close as possible to the actual post action code?

Relation to Prior Work: The presented paper fits well within the context of representation disentanglement. The authors explain clearly how their work differs from the related litarature.

Reproducibility: Yes

Additional Feedback: Many, in my opinion, important results seem to presented in the supplementary material, however, the supplementary materials archive contains only the code and a copy of the paper.


Review 3

Summary and Contributions: In this paper, the authors confirm empirically that irreducible representations are not naturally found in standard VAE models without biasing the loss landscape towards them. The authors determine that inducing such representations in VAE latent spaces garners improved performance on a number of standard disentanglement metrics. A novel disentanglement metrics to explicitly measure linear disentangled representations. Moreover, the authors propose a method to induce irreducible representations without the need for labelled action-transition pairs.

Strengths: This paper is well written and easy to read. The motivation is clear. The proposed method is novel and effective. The authors conduct extensive experiments and make this paper very solid.

Weaknesses: I do not find any obvious limitation of this this work.

Correctness: The claims and method are correct

Clarity: The paper is well-written.

Relation to Prior Work: The discussion about the differences from previous work is clear. A large number of comparative experiments have been carried out.

Reproducibility: Yes

Additional Feedback:

[Author Response · NeurIPS 2020]

We would like to thank the reviewers for their comments, especially those concerning clarity and context of our work.
Our first change will be to the paragraph titled "Convergence Consistency". The observed inconstancy was due to poor
learning rate choices for internal RGrVAE representations. We will instead focus the paragraph on general convergence
properties (including learning rate) and robustness. This alongside a reduction of Figure captions (which restated text in
places) will allow us the space to address reviewer comments. We now address the comments of each reviewer in turn.

**To Reviewer #1**    First, we thank you for your summary of related work in reinforcement learning, we hadn't considered
the context of our work in this setting. We will be adding a paragraph to our prior work section to address this.

Concerning task simplicity, we agree that work should strive to present results on datasets as close to the real world as
possible. However, we feel moving straight to noisy real world data is a large jump, especially for linear disentanglement
work where current state of the art [2] can extract up to 5 (synthetic) symmetries and required special metrics (e.g.
generative factor labels), failing when applied directly to pixels. The most extracted from pixels is 2. We intended
Section 5 to show RGrVAE performs comparably to it's supervised counterpart [1], not to show superiority. However,
we would be happy, if deemed appropriate, to present RGrVAE applied to dSprites as evidence on datasets with more
symmetries and as suggested by Reviewer #3.

Regarding experimental comparisons to RL methods, whilst we see interesting parallels between your references and
(VAE) disentanglement, it is difficult for us to determine how to appropriately compare them. This is mostly due
to little existing work bridging disentanglement in VAEs and RL, they use different datasets, different metrics and
often different (perhaps perceived) aims (ie independently controllable factors). Certainly there is space to explore
comparisons under shared methodology, but we believe this would constitute an interesting but separate work to ours.

Finally, we do agree that presenting weaknesses is important. As stated at the top of our response, we intend to replace
the consistency paragraph with one on policy convergence. It will (amongst other things) discuss robustness under
visual noise (salt+pepper, backgrounds, etc.) which, as a minor variation of the work already presented, we feel would
not require overly detailed analysis or additional figures. We will add full results to the supplementary material.

**To Reviewer #3**    We first apologise that the supplementary material was incomplete, this oversight will be rectified.

To begin we completely disagree that we have not shown evidence of linear disentanglement. Models that are not
linearly disentangled completely fail to reconstruct the post action latent and observation, please compare Tables 1 and
3a. In fact, the ability to reconstruct the latent (i.e. find an independent f) is literally the definition of linear disentangled.

Concerning the choice of number of actions, this is a hyper-parameter that (if symmetry structure is unknown) we have
to guess in the same way we guess the number of latents for VAE models (if the generative factors are unknown). The
usual approach is to allow more latents than you expect are required and the same can be done with the number of
actions. In Figure 3b we show that allowing more actions than present in the symmetry structure does not significantly
effect the policy network and thus the representation. On this note, we notice a typo in this figure where the number
of actions reported is per latent pair, not the total number available. We shall amend the figure and rephrase the
"Over representation" paragraph to be explicitly concerned with this choice. Furthermore, your suggestion of Dirichlet
processes for deciding the number of actions is intriguing, and we hope to explore this in the future.

Concerning additional tasks, we intended Section 5 to demonstrate that RGrVAE performs comparably to its supervised
counterpart [1], not to show superiority or extend it. We do however have positive preliminary results on dSprites, as
you suggested. If the reviewers think it is appropriate for the section, we would be happy to include RGrVAE action
traversals and independence measures as evidence on more complex datasets. Regarding instability of REINFORCE,
after using appropriate learning rates we found that it was extremely consistent. Furthermore, we intend to discuss
convergence under visual noise. We hope this and the possible addition of dSprites helps lessen your concerns.

We thank you for your notes on clarity, which we will address in turn. 1) This 5 is the step size (pixels) of the agent per
action. This absolutely should have been stated in the text and will be added. 2) This is the vector dot product, we will
bold vectors for additional clarity. 3) Irreducible representations of dimension 2 or higher act on 2 latent dimensions or
more, resulting in a low MIG, despite being linear disentangled. Perhaps expressive is not the correct word, we shall
rephrase to reflect that linear disentangled representations have low MIG. 4) Eq. 4 has higher reward the closer the
predicted post action latent is to the true value encouraging policies that select representations which best approximate
the action. The prediction loss $\mathcal{L}_{\text{pred}}$ (Eq. 5) encourages approximating the post action code accurately.

**To Reviewer #4**    We thank you for your review and expressing your belief in the novelty and soundness of our work.

[1] Caselles-Dupré et al. "Symmetry-Based Disentangled Representation Learning requires Interaction with Environ-
ments." NeurIPS. 2019.
[2] Pfau et al. "Disentangling by Subspace Diffusion", ArXiv. 2020


[Meta-Review · NeurIPS 2020]

In the discussion phase, reviewers expressed that, taking into account their rebuttal, authors were on a good path to have a sufficiently improved updated version for the camera ready. The AC confirms that you must include the clarifications and additions mentioned in your rebuttal (including experiments with variation with added noise/distractors and results on dSprites) in your final version. With this the work will cover sufficient interesting material, and AC and reviewers all agree that it be presented at NeurIPS.